# 'Well, it literally stops me from having a life when it's really bad': a nested qualitative interview study of patient views on the use of self-management treatments for the management of recurrent sinusitis (SNIFS trial)

Geraldine M Leydon,[1] Lisa McDermott,[2] Tammy Thomas,[1] Amy Halls,[1] Ben Holdstock-Brown,[1] Stephen Petley,[1] Clare Wiseman,[1] Paul Little,[1] on behalf of the SNIFS Investigators

GML and LMcD contributed equally.

For numbered affiliations see end of article.

**Correspondence to**
Dr Geraldine M Leydon;
gerry@soton.ac.uk

## ABSTRACT

**Objective** To explore the experience and perceptions of illness, the decision to consult a general practitioner and the use of self-management approaches for chronic or recurrent sinusitis.

**Design** Qualitative semistructured interview study.

**Setting** UK primary care.

**Participants** 32 participants who had been participating in the 'SNIFS' (Steam inhalation and Nasal Irrigation For recurrent Sinusitis) trial in the South of England.

**Method** Thematic analysis of semistructured telephone interviews.

**Results** Participants often reported dramatic impact on both activities and their quality of life. Participants were aware of both antibiotic side effects and resistance, but if they had previously been prescribed antibiotics, many patients believed that they would be necessary for the future treatment of sinusitis. Participants used self-help treatments for short and limited periods of time only. In the context of the trial, steam inhalation used for recurrent sinusitis was described as acceptable but is seen as having limited effectiveness. Nasal irrigation was viewed as acceptable and beneficial by more patients. However, some participants reported that they would not use the treatment again due to the uncomfortable side effects they experienced, which outweighed any symptom relief, which may have resulted had they continued.

**Conclusions** Steam inhalation is acceptable but seen as having limited effectiveness. Nasal irrigation is generally acceptable and beneficial for symptoms, but detailed information on the correct procedure and potential benefits of persisting may increase acceptability and adherence in those patients who find it uncomfortable.

**Trial registration number** ISRCTN 88204146.

## Strengths and limitations of this study

► The interviews permitted an exploration of participants' perceptions of particular management strategies for recurrent or chronic sinusitis.

► Participants represented a sample of patients with recurrent sinusitis who had managed their condition using different treatment options.

► Trial participation may have led to a sample of participants particularly interested in this research and therefore may not be representative of 'typical' patients.

► Due to study procedure limitations, the team was unable to collect characteristics of all participants and this has reduced our ability to analyse data according to key characteristics, such as trial arm, in comparative ways.

prescriptions for RTIs account for over 60% of all antibiotic prescriptions in primary care.[1] One study of general practice found antibiotics to be prescribed in around 54% of all consultations for RTIs.[2] Despite the large number of antibiotics frequently prescribed for RTIs, evidence has consistently demonstrated the limited benefit of antibiotics in treating these conditions.[1 3 4] Furthermore, the overuse of antibiotics can contribute to the spread of resistant bacteria.[5 6] This problem is currently on the increase and has been identified by WHO as a serious issue that must be addressed with urgency.[7]

While antibiotics may be beneficial for some patients with sinusitis, the National Institute for Health and Care Excellence guidelines recommend that antibiotics should not be prescribed for RTIs in most instances, unless

## BACKGROUND

Many respiratory tract infections (RTIs) are acute and self-limiting conditions, lasting for around 1–2 weeks. However, antibiotic

**Table 1** SNIFS trial advice strategies

| Advice strategy | Definition |
| --- | --- |
| Nasal saline irrigation | Participants given verbal instructions and a link to a demonstration video on YouTube. Participants provided with a neti pot and asked to irrigate their nose daily for 6 months. |
| Steam inhalation | Participants were asked to inhale steam for 5 min each day. |
| Usual care | Use of medications or referral was at discretion of patient's physician. |

a patient meets a specific at-risk criteria, such as being systematically unwell, or at high risk of serious complications due to a pre-existing comorbidity.[1] However, a number of 'self-help' treatments can be recommended to assist in relieving symptoms related to RTIs.[1] In relation to sinusitis (which is classified as an RTI), there is some evidence to show that both steam inhalation (involving a patient inhaling steam over a bowl of boiling water) and nasal irrigation (the irrigation of nasal cavities with saline) can reduce symptoms. However, the evidence is inconsistent and limited.[8–10]

Previous qualitative work among patients with chronic sinusitis in secondary care suggests significant impact on quality of life and dissatisfaction with treatments received.[11] However, most recurrent or chronic sinusitis is managed in primary care, and we are aware of no qualitative studies to explore patients' perceptions in primary care. The SNIFS (Steam inhalation and Nasal Irrigation For recurrent Sinusitis) trial aimed to assess the efficacy of steam inhalation and nasal irrigation for the treatment of sinusitis. The study recruited 871 patients from across 72 practices in the South of England. Participants were randomly assigned to an advice strategy in a 2×2 factorial design: to receive advice to use nasal saline irrigation daily or no such advice; each of these groups was also randomised to receive advice to use steam inhalation daily or no such advice. This advice is shown in table 1. All study participants had access to usual care. Participants had had at least two episodes of acute sinusitis or one episode of chronic sinusitis (lasting for 12 weeks or more) in the 3 years before enrolment.[12]

The main findings from the trial have been published and demonstrate modest benefit at 3 and 6 months from nasal irrigation but not for steam inhalation.[12] This paper reports on the findings of a nested qualitative study that aimed to explore the feasibility and acceptability of these treatments from the patient perspective, the experience of illness and previous treatments and factors that influence patient decisions to consult a general practitioner (GP) or use self-help treatments (including steam inhalation and nasal irrigation) for the symptoms of sinusitis.

## METHODS
### Participants and procedure
Participants were recruited from patients with chronic or recurrent sinusitis who were taking part in the SNIFS trial. Patients had seen the GP for previous episodes of sinusitis and were also experiencing symptoms currently, although they did not have any 'at-risk criteria' as this would have excluded them from participating in the trial. Participants in the trial consented to being invited by telephone for an interview. Participants who had withdrawn from the trial were also eligible to take part in the interviews. The participants were recruited from areas across Southampton and Hampshire. Interview participants were purposively sampled to include a range of participants according to their allocation in the trial (eg, steam inhalation/nasal irrigation) and age and gender.

### Interviews
Trained interviewers LMcD and CW (female) and SP and BH (male) conducted telephone interviews (in order to include a wide geographical area), with each lasting approximately half an hour. Ethical approval was in place for interviews to last up to 60 min and no interviews exceeded this. SP, BH and CW were medical students during their involvement in this research and supervised by senior academics (GL and PL). Each participant was interviewed once, and all interviews were audio-recorded and transcribed verbatim in preparation for analysis. Semistructured qualitative interviews allowed researchers to gather insights into participants' individual views and experiences of treatments for sinusitis, as well as providing a structure for comparison across different accounts.[13] The interview guide (online supplementary appendix 1) was developed as part of a student project and tested with a member of staff: this provided a training opportunity for interviewing and feedback on functionality. Previous literature was reviewed and questions thought to be relevant included. It was then refined and agreed by the research team to ensure our aims were met. It included key topic areas while also providing flexibility to explore unanticipated issues. Subtle realism best characterises our epistemological position.[14]

### Analysis
Inductive thematic analysis[15] was conducted on all transcripts to determine factors that influence patients' decision to consult a GP or use an alternative treatment for sinusitis, as well as being open to themes outside of the core aims of the study. Following immersion in the transcripts, familiarisation was achieved and recurrent patterns and prominent themes were identified and labelled with codes. Each code label referred directly to the operationalisation of the theme content. A label and full descriptive definition was then provided for each theme. The codes and definitions

**Table 2** Themes identified in analysis

| Themes | Subcategories |
| --- | --- |
| 1. Perceptions of severity | -Duration of symptoms<br>-Perceived signs of severity<br>-Quality of life impact |
| 2. Advice from others | -Acceptance of GP advice<br>-Consideration of alternative advice |
| 3. Perceptions of antibiotics | -Antibiotics have unpleasant side effects<br>-Concerns about resistance<br>-Previous experience of antibiotics |
| 4. Perceptions of self-help treatment | -Previous experience of treatments<br>-Treatment duration short and irregular |
| 5. Experiences of steam inhalation | -Beneficial but only for short-term relief<br>-Beneficial but only for severe symptoms |
| 6. Experiences of nasal irrigation | -Strong benefits of treatment outweigh any discomfort<br>-Uncomfortable side effects of treatment cannot justify use |

GP, general practitioner.

were refined during a continuing process, which involved themes being linked, grouped, moved, relabelled, added and removed to produce a set of themes and subthemes and a coding manual, which adequately fitted and thoroughly explained the data. The coding was iteratively developed across lead authors (led by LMcD and GL) and adjustments made where appropriate based on team discussion. Data saturation[16] was achieved and recruitment ceased, with no further interviews conducted.

## FINDINGS
### Participants
In total, 32 participants took part in the study. The age of participants ranged from 18 to 74 (mean age 55). Approximately 72% (23) were women and 28% (9) men.

### Themes
Thematic analysis identified a total of six themes relating to using self-help treatments (steam inhalation/nasal irrigation/other remedies) or consulting with a GP for the treatment of sinusitis. Illustrative quotations are provided and details relating to the interviewee are provided in parentheses. The themes are shown in table 2.

### 1. Perceptions of severity
Perceptions of severity were shaped by the duration of symptoms, perceived indicators of severity and particularly by the impact of the illness on quality of life.

#### Duration of symptoms
Most participants reported that an evaluation of severity was based on the number of days that they had experienced symptoms, which they related to the current attack (this ranged from a few days to weeks).

*It was horrible, I tried to cope for about a week on my own but in the end I just had to go (to the GP) (Participant L05).*

#### Perceived signs of severity
A number of factors were used as indicators of severity including a variety of signs and symptoms ranging from pain (head, sinuses, face) to pressure (eg, nasal passages being blocked), noises (from nose) and sensations (around face and head).

*It's just the… like my face hurting, my headaches and just the… lethargic and you know just everything about it really (Participant L04).*

#### Quality of life impact
Participants discussed the way in which their symptoms had prevented them from attending work, caring for their children and taking part in social or other activities.

*Well, I don't tend to go swimming or anything like that, or a lot of exercise. And even gardening, things where I have to put my head forward, you know, my brain feels like it's bouncing inside my head. So I don't tend to do that sort of thing (Participant B03).*

Others described the sinusitis as having an impact on life in general.

*Well, it literally stops me from having a life when it's really bad because I really can't get up and walk about. So it does interfere with my life (Participant B10).*

### 2. Advice from others
#### Acceptance of GP advice
Overwhelmingly, most participants reported that they were happy to accept and follow their GP's treatment advice. This advice could include taking antibiotics or using various self-help treatments such as nasal irrigation or steam inhalation.

*If I was advised to (use steam inhalation), yes. I mean, it's probably not something I would think about just getting on and doing, but if I was asked—if a GP suggested it then, yes I would (Participant L06).*

### Consideration of alternative advice
Sources other than their GP were commonly used in decision making, predominantly family members, but also the internet and newspapers.

*We had relatives over from Australia who brought me a bottle (of eucalyptus oil to inhale) and that's fantastic, that's for a general cold as well (Participant L05).*

### 3. Perceptions of antibiotics
Individuals' perceptions of antibiotics were related to beliefs about side effects, concerns about resistance and their previous experience (good or bad) of taking antibiotics.

### Antibiotics have unpleasant side effects
Many participants were well aware of side effects, mainly related to stomach complaints, although could include a variety of effects such as skin rashes or mouth ulcers.

*It can give you an upset stomach, well I've experienced it myself, because doesn't it destroy the good bacteria or something? (Participant B08).*

*Well… they (antibiotics) upset my stomach sometimes. (Participant B10).*

*I think some people I've met can't take them, they come out in quite a severe rash sometimes, and I have noticed, perhaps like tiny ulcers appearing in my mouth, which I've found does happen with antibiotics sometimes (Participant L02).*

### Concerns about resistance
Many participants were also aware of, and worried about, the issue of antibiotic resistance but commonly described it as the body becoming resistant.

*We're building up a resistance to them and everything. So I do know there is a problem using them. I felt quite worried when I had to take 2, 100 milligrams in a week (Participant B10).*

*I know it's not the answer because we're all getting immune to them… I am very aware that we can get, well, you know, resistant to antibiotics (Participant B05).*

### Previous experience of antibiotics
Patients who attributed symptom resolution specifically to antibiotics believed they would be effective and might want to take them again for the same symptoms.

*Well, only the antibiotics, was the only thing that got rid of it completely (Participant B04).*

*Very often the only thing I can have is an antibiotic to cure it (Participant B05).*

*I've got to have antibiotics (Participant B05)*

*Well, I got to the point where I just used to ring them up and say, "I know what I've got—can I have some antibiotics please?" (Participant B04).*

### 4. Perceptions of self-help treatment
### Previous experience of treatments
Previous experience of using self-help treatments also strongly influenced their decision of whether or not to use them again, to consult a GP or to try other self-help methods.

*Done it (steam inhalation) all my life over the years, so no, no, not a problem doing things like that. It's the way—to be buried under a towel… (Participant L07).*

*It's (inhalation) something I've done for quite a few years with them (Participant B03).*

### Treatment duration short and irregular
Some participants reported that, in general, they used self-help treatments only for short and limited periods of time. Self-help treatments appeared to be used in an irregular and inconsistent way (eg, stopping treatment (such as steam inhalation) as soon as relief is first experienced).

*Sometimes I think perhaps I've—that is excessive (using nasal irrigation twice a day), especially now that I feel everything is clearer, but particularly to start with, that was quite a relief (Participant B06).*

*Well I've never used it (steam inhalation) more than probably half a dozen times for one session (illness) over a period of days (Participant L09).*

### 5. Experiences of steam inhalation
### Beneficial but only for short-term relief
Most participants who had experienced steam inhalation—either as part of the trial, following previous advice from a GP, or as an independent technique that the participant had previously tried—reported that the treatment could be beneficial in relieving symptoms. However, accounts signalled a belief that symptoms were only reduced for a short period of time (up to a few hours).

*That steam inhalation does actually provide immediate— it clears the airways, but, again, it just doesn't last. Even if I do it on a regular basis, it's like—you know, within a very short period after doing the inhalation I'm blocked up again… within half an hour (Participant L05).*

Some participants who had experienced steam inhalation reported that they would only use it if their symptoms were significantly disruptive, positioning steam inhalation as a self-management technique as perhaps better suited for more severe symptoms.

*I haven't used it (steam inhalation) this time, I suppose it's (sinusitis) not bad enough to use that, but if I couldn't sleep and that, then I would have to do that (inhale steam) (Participant L04).*

*I only do it (steam inhalation) if I'm feeling really bad, if it's*

*getting really bad (Participant L08)*

### 6. Experiences of nasal irrigation

Participants' experiences of using nasal irrigation also influenced their decisions to use the method again or whether to consult a GP. In similar fashion to steam inhalation, the treatment of nasal irrigation had been experienced by participants either as part of the trial, but for some following previous advice from a GP, or as an independent technique that the participant had tried previously. Unsurprisingly, it was clear that the balance of discomfort and symptomatic benefit influenced participants' use of irrigation.

*Strong benefits of treatment outweigh any discomfort*

Some of the participants who had experienced nasal irrigation reported that the technique provided relief for their symptoms and that although the treatment may be slightly uncomfortable at times, the benefits received outweighed any discomfort suffered.

> *I can put up with it because it does improve the sinus condition (Participant C01).*

> *After the initial shock of having to do it, it does help. It helped relieve symptoms… It freaked me out a bit to start with. But, once you get into the hang of it, it's alright (Participant B09).*

*Uncomfortable side effects of treatment cannot justify use*

However, around half of the participants who experienced nasal irrigation reported that the treatment was too uncomfortable or unpleasant. Side effects reported included: shuddering, an increase in the feeling of mucus being produced, water in the back of the throat, water running out of the nose hours later and general pain.

> *One of my memories is of the actual shuddering and in the end I just gave up because I found it very unpleasant to do (Participant L01).*

> *It doesn't actually drain it, you can actually blow your nose and get rid of it. It goes everywhere. It goes down the back of your throat and you end up coughing (Participant B01).*

> *I don't know whether it's the configuration of the sinuses or whatever, but instead of, sort of, coming out the other side, it was going straight down the tubes into my ears. And, well, that was quite uncomfortable (Participant B03).*

### DISCUSSION

We are aware of no prior qualitative studies of patients' perceptions of steam inhalation and nasal irrigation for recurrent or chronic sinusitis in a primary care setting. The study identified six key themes related to factors that influence patients' decisions to consult a GP or use a self-help treatment (in particular steam inhalation or nasal irrigation) for symptoms of sinusitis. Findings are discussed in relation to the most prevalent and influential themes outlined by participants.

### MAIN FINDINGS

Patients with recurrent or chronic sinusitis described the often dramatic impact on their activities and quality of life and viewed their sinusitis as a chronic condition. They based most treatment decisions on past experiences of managing symptoms, although many were willing to accept GP advice. Thus, although many participants were well aware of the potential negative consequences of antibiotics and some did not expect to receive them every time they consulted a GP, if they had previously been prescribed antibiotics, they often believed they would be necessary for the future treatment of sinusitis. Patients generally used self-help treatments for short and limited periods of time only. Interviewees viewed steam inhalation as acceptable and beneficial for symptoms. It was commonly perceived as a method that was only beneficial in providing short-term relief, and some participants believed it was mostly helpful for severe symptoms. Similarly, nasal irrigation was generally viewed by interviewees as acceptable and beneficial for symptoms, although some would not use the treatment again due to the balance of uncomfortable or unpleasant side effects, which outweighed any potential symptom relief. However, within the context of the SNIFS trial, detailed information on the correct procedure and potential benefits of both treatments may have helped to increase patient acceptability and adherence in those patients who experienced discomfort.

### COMPARISON WITH EXISTING LITERATURE

If participants had been prescribed antibiotics in the past for the treatment of sinusitis, they attributed symptom benefit to antibiotics and accounts indicated beliefs that this treatment would, therefore, be the optimal approach for future management of the condition. In addition, participants also reported an acceptance of any GP advice, suggesting that, if GPs prescribe antibiotics for sinusitis, patients' beliefs that antibiotics may be needed in the future will be strongly reinforced. These findings are supported by previous quantitative research, which has documented that prescribing antibiotics for RTIs directly influences patients' views relating to the need to consult a GP and take antibiotics for future RTIs.[17]

One factor maintaining high prescription rates for sinusitis (with more than 90% of individuals receiving antibiotics[2]) is likely to be the significant impact of the condition on the patient's quality of life—described in this study in such dramatic terms as 'horrible' and 'it literally stops me from having a life'. The significance of impact on quality of life is supported by similar findings from a secondary care sample.[11] A further factor is that sinusitis is second only to chest infections in a long duration of each attack—on average just short of 3 weeks.[17] Thus, given an unpleasant and long-lasting condition, if GPs have nothing else to recommend, antibiotics are likely to be used.

However, patients were aware of the consequences of antibiotics and many would be happy to accept GP advice

not to take them, if this was recommended. This lack of expectation for antibiotics has also been reported in research with patients with lower RTI and sore throat and in particular often contrasts with GP perceptions of high levels of patient pressure to prescribe (eg, see Braun and Clarke, Thompson *et al*[15 18]). Therefore, the findings suggest that, if GPs could reduce their prescribing of antibiotics for sinusitis, many patients will be happy to accept this.

## IMPLICATIONS OF USING STEAM INHALATION AND NASAL IRRIGATION IN CLINICAL PRACTICE

Overall, patients reported that steam inhalation was an acceptable treatment for the symptoms of sinusitis. However, many patients believed that steam inhalation was only beneficial for the short-term relief of symptoms and this corresponds with findings from the main clinical trial, which found limited benefit.[12] While clearly some patients will feel they get benefit from steam inhalation, there is also some evidence from other studies of mild thermal injury using steam.[19] Overall, the combined qualitative and quantitative findings suggest that there is a limited place for routinely advising patients to use steam inhalation for chronic or recurrent sinusitis.

Patients who had experienced nasal irrigation could be identified in one of two subgroups: those who described irrigation as an acceptable technique that could relieve their symptoms and those who described the discomfort experienced during irrigation with limited justification for its use as a treatment. Patients who reportedly found irrigation to be acceptable often reported that they had persisted with it despite initially finding the treatment very uncomfortable, and many reported how they had 'got used to' the discomfort in order to experience the benefits. Therefore, it is possible that the patients who found irrigation unacceptable may have changed their mind if they had persisted with the treatment. In addition, patients who reported irrigation as being unacceptable tended to report side effects such as water going down their throat. In short, findings suggest that nasal irrigation can be viewed as an acceptable treatment for the symptoms of sinusitis. An important caveat to this, however, is the need for detailed and clear patient information on the correct procedure and the potential benefits of persisting with the technique in terms of finding it easier with increased practice/use of nasal irrigation. Clinicians are recommended to give greater support to patients in using the technique, given the effect shown in the trial (modest benefit at 3 and 6 months from nasal irrigation).

## STRENGTHS AND LIMITATIONS

One of the key strengths of the study lies in the fact that all patients interviewed had experienced at least one of the treatment options (steam inhalation or nasal irrigation) as part of the SNIFS trial. Therefore, the interviewees could discuss in-depth the various treatment options in accordance with the way in which the treatments were delivered during the trial.

However, the sample of participants who took part in the interviews may have limited the scope of the findings due to the fact that they had all consented to and taken part in the SNIFS trial. Trial participation may have led to a sample of participants who were particularly interested in research of this nature and may not have represented views held by non-trial patients. In particular, participants may have held more positive views towards self-management treatments due to their interest and participation in research of this nature.

Despite this possible 'research-biased' sample, all patients interviewed did represent a sample of patients who had experienced recurrent sinusitis and had also managed their condition using a number of treatment options over the years. In addition, a few patients interviewed had also withdrawn from the trial but were happy to take part in an interview, suggesting that a wider range of patient views were included within the sample (and not simply those who may view self-management treatments in a more positive light).

## CONCLUSION

The findings from this qualitative study suggest that steam inhalation is viewed as an acceptable treatment that patients are happy to use, although many perceive it as having limited short-term benefit. Nasal irrigation is acceptable to many patients in relieving symptoms, but some find it uncomfortable or mildly unpleasant. However, detailed information on the correct procedure and potential benefits of persisting with the technique may increase the acceptability of nasal irrigation in patients who find it initially uncomfortable, a finding supported by the main trial data.

**Author affiliations**
[1]Primary Care and Population Sciences, Faculty of Medicine, University of Southampton, Aldermoor Health Centre, Aldermoor Close, Southampton, SO16 5ST, UK
[2]Primary Care and Public Health Sciences, King's College, 614, 6th Floor, Capital House, 42 Weston Street, London, SE1 3QD, UK

**Acknowledgements** The authors are grateful to the patients and health care professionals who contributed their time and effort and helpful insights to make the study possible. They are also grateful to the Programme Steering Committee for their support and advice throughout the study, National Institute for Health Research for funding Geraldine Leydon on a personal fellowship during this study and C Smith who contributed to the research while a medical student.

**Collaborators** SNIFS Investigators: Universityof Southampton: Pat Alexant, Sophie Johnson, Chris Smith.

**Contributors** GML led the qualitative work as part of the SNIFS trial. PL secured research funding and acted as overall PI of the SNIFS trial. TT facilitated coordination and recruitment with P Alexant and S Johnson. SP, BH-B, C Smith and CW collected interview data. All authors were involved in/commented on data analysis (led by GML and LMcD). AH progressed the manuscript and all authors contributed to its writing.

**Funding** The study was funded by a grant from the National Institute for Health Research (NIHR) under its Programme Grants for Applied Research (grant no.

RP-PG-0407-10098). The views expressed are those of the authors and not necessarily those of the National Health Service, NIHR or the Department of Health. The University of Southampton was the sponsor, but it had no role in the running of the study, the analysis or interpretation of the results or the preparation of the manuscript.

**Competing interests**  None declared.

**Ethics approval**  This study was given ethical approval by the Hampshire Rec B Research Ethics Committee (number 07/Q1704/69).

**Provenance and peer review**  Not commissioned; externally peer reviewed.

**Data sharing statement**  No additional data are available

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
