## [Reviewer comments · BMJ Open]

ARTICLE DETAILS

TITLE (PROVISIONAL)	'Well, it literally stops me from having a life when it's really bad' : a nested qualitative interview study of patient views on the use of self-management treatments for the management of recurrent sinusitis (SNIFS trial).
AUTHORS	Leydon, Gerry; McDermott, Lisa; Thomas, Tammy; Halls, Amy; Holdstock-Brown, Ben; Petley, Stephen; Wiseman, Clare; Little, Paul

VERSION 1 - REVIEW

REVIEWER	John Campbell University of Exeter
REVIEW RETURNED	09-Apr-2017

GENERAL COMMENTS	Thank you for the opportunity to review this qualitative paper from the Southampton Primary Care Team. • The abstract is generally acceptable.• Some clarification would be helpful in 2/17 and 2/36 – are these observations made directly in relation to the trial i.e. “steam inhalation used for recurrent sinusitis is acceptable, but is seen as having limited effectiveness” (note grammar, punctuation).• Clairfication is need in 2/49 -what exactly is “novel”• 2/55 - delete “despite this” – these items are independent clause and lists.• 3/4 – what does “relatively brief” means – be specific – relative to what?• 3/7 – requires rephrasing – what does it mean to say “the prescribing of antibiotics for RTIs accounts for 54% of all consultations for RTIs” – does this mean that “during the course of consultations for RTIs, 54% of consultations result in prescription of an antibiotic”? Please clarify.• Overall, some attention to grammar and presentation is necessary – in 3/14 for example ‘... increase and which has been identified ...’• 4/4 – were participation recruited from a sample of patients or “... patients with...”?• 4/25 – what is the basis of the suggestion that qualitative interviews are the “best” method....• 4/27 - how was the interview guide developed?• 4/32 – the analysis section is clear and well described.• Page 6 – there is no reference to monitoring of temperature – did this not come out in the interviews?• 8/55 – including the prepositions will add clarity when linking clauses in sentences – “whether or not to use them again, to consult a GP, or to try other self-help methods”• 9/14-15 and 10/24 – punctuation.• 9/19 – missing)• 12/27 – it is not clear to me why the “medicalisation of illness cannot easily explain ...” – what is the rationale?
---

	 • 12/35 – “on”, not “of” quality of life • 12/37 – “another” should be “a further factor”. • 12/50 – represents a council of despair – the key issue the reader wants to see is how GPs can address the issue of their prescribing, not a vague statement about the potential to do so. • 12/44 – is this sentence actually correct? – I doubt it. I suspect that some patients will be happy, but others will not be happy. • 13/57 – I don’t follow the logic for the assertion that “this strengthens the findings” – can this be clarified? • 14/12 – the authors haven’t identified what a “typical” patient is – to what extent are study participants from this study representative of such patients? • 15/39 – for NIHR funded work I don’t understand why there are no data sets available for sharing. I thought that this was an NHIR requirement and the availability of qualitative data would be potentially valuable to other researchers interested in the area. • 17 – the COREQ checklist is substantially incomplete. Despite these observations, I found this to be an interesting and relevant article. It addresses an important area of clinical practice and the qualitative material adds depth to the findings of the previously published trial. Thank you for the opportunity to review it.
--	---

REVIEWER	Sarah Tonkin-Crine University of Oxford, UK I have previously worked with two of these authors on projects related to antibiotics and respiratory tract infections but have no previous knowledge or connection with the current study presented or the wider programme grant.
REVIEW RETURNED	11-Apr-2017

GENERAL COMMENTS	This manuscript presents an important and novel study of participant experiences of using self-management strategies for the management of chronic sinusitis. The manuscript is well written and presents a high quality study however some sections of the manuscript could be amended to better clarify the main findings. Title  1. The quote presented in the title is about the experience of sinusitis whereas the main aim of this study was to explore experiences of self-management techniques. 2. The title also mentions participants’ decision to consult a GP for sinusitis which again is not the main aim of the manuscript (very little in results relating to this). Abstract  1. The results and discussion in the abstract do not appear to accurately reflect the results presented, see later comments. 2. It should be made clear in the discussion which are views of the participants and which are findings from the main trial. 3. The first strength listed under strengths and limitations is not very clear. I would have thought the novelty of the study was exploring perceptions of particular management strategies? But if it is patients being managed in primary care this could be re-worded. Background  1. It would be helpful to clarify that antibiotics may be beneficial for some patients with (rhino)sinusitis and give examples of “at risk
--

criteria” mentioned in the NICE guidelines.

2. It would be useful to have definitions of recurrent and chronic sinusitis for a non-clinical reader to distinguish whether these are substantially different.

3. It would be useful to specify what participants were required to do as part of the trial. How long were they trying the self-management treatments for? What instructions were they given about using treatments? Did any patients get antibiotics as well? Did all patients only try one of the two strategies? (e.g. no patient had experience of both?)

Methods

1. Recruitment should mention that participants who had withdrawn from the trial were also eligible to take part in interviews.

2. Exact interview lengths should be reported under the results.

3. It would be useful to include a copy of the interview guide to compare this to the themes identified.

4. The description of identifying themes and codes in the analysis does not sound like an inductive process as it sounds like themes were identified straight away?

5. Ethical approval is not reported.

Results

1. Results do not report the response rate for participants invited to take part in the study.

2. Participant characteristics should include the trial arm to which the participant was randomised and whether they completed trial activities (some withdrew – reasons for this should be noted).

3. Results do not specify whether participants had recurrent or chronic sinusitis (if these are different) and how long they had had the condition. It would also be useful to know how many had treated their symptoms with antibiotics before and how many had used self-management outside of the trial.

4. Did any of these participants have specific “at risk criteria” mentioned in the background which would indicate that antibiotics may be useful for treating their condition – presumably these people were excluded from the trial? Clarify this to establish that these patients can be treated with antibiotics.

5. It is not clear why some words within quotes are presented in bold text.

6. The subtheme “acceptance of GPs advice” is important and suggests that patients are happy to try self-management strategies if their GP thinks it would help. The acceptance of non-antibiotic strategies following previous experience of antibiotic treatment and beliefs about the effectiveness of antibiotics for sinusitis is unclear and more evidence under this theme would be helpful. At present these results contrast with the subtheme “previous experience of antibiotics”.

7. The subtheme “beneficial but only for severe symptoms” does not seem to be supported by the quotes. The quotes indicate that participants would not bother to try steam inhalation unless their symptoms were “bad” indicating that steam inhalation is seen as effortful and only worth trying if symptoms are disruptive? Neither quote talks about perceived effectiveness.

Discussion

1. The results do not support the statement that patients viewed their sinusitis as chronic – need to provide evidence of this or else delete this sentence.

2. Authors report that patients based treatment decisions on past

	experience – but this counteracts the subtheme about accepting GP advice. Again it is not clear how perceptions of antibiotic effectiveness and previous experience of antibiotics influenced perceptions of using self-management strategies. 3. Line 44, reports that patients would be happy not to take antibiotics following GPs advice but evidence for this is not supported in the results. 4. The section on comparison with existing literature could be condensed to the main points relevant to the results of this study. 5. Implications regarding steam inhalation could be more clearly presented with appropriate weight given to the trial results. Implication for nasal irrigation could be shorter (avoid repeating the results) and it would be helpful to clarify that clinicians are recommended to give greater support to patients in using the technique given the effect shown in the trial. 6. In the conclusion, again specify what is a result of the trial and what is participants' views. 7. Boxes on the COREC form appear to be blank – these should be reported in the manuscript or else have N/A next to them. 8. The four authors listed as undertaking interviews in the methods section do not match the initials reported under contributorship.
--	--

REVIEWER	Kaatje Van Roy Department of Family Medicine and Primary Health Care Faculty of Medicine Ghent University Belgium
REVIEW RETURNED	12-Apr-2017

GENERAL COMMENTS	congratulations with this clear and interesting article! I only have some very minor remarks related to the questions above:  - I can't see any information on the ethics approval. It is probably part of the ethics approval for the larger trial, but this is not made explicit. - I can't find information on the number/motivations of participants who declined participation in the telephone interviews. - Document COREQ seems to be incomplete. Related to the text:  - In the manuscript, p 6, the term 'feelings' is maybe a bit strange, as it generally refers to psychological feelings or emotions. Could the example given be classified under 'sensations'? - The title refers to 'the decision to consult a GP': I am not sure whether this is really the focus of the results. - It would be helpful to attach the interview guide as an appendix
--

VERSION 1 – AUTHOR RESPONSE

Reviewer 1

Comment:

The abstract is generally acceptable.

Response:

Thank you.

Comment:

Some clarification would be helpful in 2/17 and 2/36 – are these observations made directly in relation to the trial i.e. “steam inhalation used for recurrent sinusitis is acceptable, but is seen as having limited effectiveness” (note grammar, punctuation).

Response:

This has been edited on page 2 and now reads as follows: “In the context of the trial, steam inhalation used for recurrent sinusitis was described as acceptable, but is seen as having limited effectiveness.”

Comment:

Clarification is need in 2/49 -what exactly is “novel”

Response:

Thank you for this observation. It has been edited as follows on page 2 and hopefully is now clearer: “The interviews permitted an exploration of participants’ perceptions of particular management strategies for recurrent or chronic sinusitis.”

Comment:

2/55 - delete “despite this” – these items are independent clause and lists.

Response:

Thank you, this has been deleted as suggested.

Comment:

3/4 – what does “relatively brief” means – be specific – relative to what?

Response:

This sentence has been edited on page 3 and now reads as follows: “Respiratory tract infections (RTIs) are self-limiting conditions, lasting around one to two weeks.”

The following two comments have been addressed together:

3/7 – requires rephrasing – what does it mean to say “the prescribing of antibiotics for RTIs

and:

accounts for 54% of all consultations for RTIs” – does this mean that “during the course of consultations for RTIs, 54% of consultations result in prescription of an antibiotic”? Please clarify.

Response:

Thank you for these comments, we have combined these sentences on page 3 and they now read as follows: “Respiratory tract infections (RTIs) are self-limiting conditions, lasting around one to two weeks. However antibiotic prescriptions for RTIs account for over 60% of all antibiotic prescriptions in primary care[1]. One study of general practice prescribing found antibiotics to be prescribed in around 54% of all consultations for RTIs[2].”

Comment:

Overall, some attention to grammar and presentation is necessary – in 3/14 for example ‘... increase and which has been identified’

Response:

Thank you for this comment, the sentence has been edited on page 3 and now reads as follows: “Furthermore, the overuse of antibiotics can contribute to the spread of resistant bacteria[5,6]. This problem is currently on the increase and has been identified by the World Health Organisation as a serious issue which must be addressed with urgency[7].”

Comment:

4/4 – were participants recruited from a sample of patients or “... patients with...”?

Response:

Thank you for the comment, this has been edited on page 4 and now reads as follows: “Participants were recruited from patients with chronic or recurrent sinusitis who were taking part in the SNIFS trial.”

Comment:

4/25 – what is the basis of the suggestion that qualitative interviews are the “best” method....

Response:

Thank you for this question. We have edited this sentence on page 5 as follows: “This sentence has been edited for clarity and now reads as follows: “Semi-structured qualitative interviews allowed researchers to gather insights into participants’ individual views and experiences of treatments for sinusitis, as well as providing a structure for comparison across different accounts[13].”

Comment:

4/27 - how was the interview guide developed?

Response:

Thank you for this comment, we agree we should have provided more information. We have edited this section on page 5 and it now reads as follows: “The interview guide was developed by the research team to ensure our aims were met. Previous literature was reviewed and questions thought to be relevant included. The guide was tested, refined and agreed. It included key topic areas while also providing flexibility to explore unanticipated issues.”

Comment:

4/32 – the analysis section is clear and well described.

Response:

Thank you.

Comment:

Page 6 – there is no reference to monitoring of temperature – did this not come out in the interviews?

Response:

We have read back through the interview transcripts and monitoring temperature is not referred to.

Comment:

8/55 – including the prepositions will add clarity when linking clauses in sentences – “whether or not to use them again, to consult a GP, or to try other self-help methods”

Response:

Thank you, this sentence has been amended as suggested.

Comment:

9/14-15 and 10/24 – punctuation.

Response:

Thank you for these observations.

9/14-15 this sentence has been edited on page 11 and now reads as follows: “Some participants reported that in general they used self-help treatments only for short and limited periods of time, judging regular and consistent use as unnecessary”

10/24 this sentence has now been edited for clarity on page 11 and reads as follows: “Unsurprisingly, it was clear that the balance of discomfort and symptomatic benefit influenced participants’ use of irrigation”

Comment:

9/19 – missing)

Response:

Thank you. This has been added in.

Comment:

12/27 – it is not clear to me why the “medicalisation of illness cannot easily explain ...” – what is the rationale?

Response:

This section has been deleted, in line with comments from Reviewer 2.

Comment:

12/35 – “on”, not “of” quality of life

Response:

This has been amended as suggested.

Comment:

12/37 – “another” should be “a further factor”.

Response:

This has been amended as suggested.

Comment:

12/50 – represents a council of despair – the key issue the reader wants to see is how GPs can address the issue of their prescribing, not a vague statement about the potential to do so.

Response:

Thank you for this comment, it has been edited on page 14 and we hope is now clearer: “Therefore, the findings suggest that if GPs could reduce their prescribing of antibiotics for sinusitis, many patients will be happy to accept this.”

Comment:

12/44 – is this sentence actually correct? – I doubt it. I suspect that some patients will be happy, but others will not be happy.

Response:

Thank you for this question. We have edited this on page 14 as follows: “However, patients were aware of the limitations of antibiotics and many would be happy to accept GP advice not to take them, if this was recommended.”

Comment:

13/57 – I don’t follow the logic for the assertion that “this strengthens the findings” – can this be clarified?

Response:

Thank you for the comment, we have deleted this last sentence.

Comment:

14/12 – the authors haven't identified what a "typical" patient is – to what extent are study participants from this study representative of such patients?

Response:

We agree this was not very clear. As such, this last sentence has been deleted.

Comment:

15/39 – for NIHR funded work I don't understand why there are no data sets available for sharing. I thought that this was an NHIR requirement and the availability of qualitative data would be potentially valuable to other researchers interested in the area.

Response:

As far as we are aware, NIHR policy relates currently to quantitative data. We do not have data available for sharing due to issues of confidentiality.

Comment:

17 – the COREQ checklist is substantially incomplete.

Response:

We have uploaded an edited COREQ checklist as part of the resubmission.

Comment:

Despite these observations, I found this to be an interesting and relevant article. It addresses an important area of clinical practice and the qualitative material adds depth to the findings of the previously published trial. Thank you for the opportunity to review it.

Response:

Thank you very much for your positive review and very helpful comments and suggestions.

Reviewer 2

Comment:

This manuscript presents an important and novel study of participant experiences of using self-management strategies for the management of chronic sinusitis. The manuscript is well written and presents a high quality study however some sections of the manuscript could be amended to better clarify the main findings.

Response:

Thank you. We are pleased that you find it a well-written presentation of a high quality study. We hope we have successfully addressed your comments below.

The following two comments have been addressed together:

Title

1. The quote presented in the title is about the experience of sinusitis whereas the main aim of this study was to explore experiences of self-management techniques
2. The title also mentions participants' decision to consult a GP for sinusitis which again is not the main aim of the manuscript (very little in results relating to this).

We understand the title may have been misleading. We have edited it to read: "Well, it literally stops me from having a life when it's really bad': a nested qualitative interview study of patient views on the use of self-management treatments for the management of recurrent sinusitis (SNIFS trial).' The quote gives context on the impact of sinusitis whilst then focusing in on consulting/self-management for the condition.

Comment:

Abstract

1. The results and discussion in the abstract do not appear to accurately reflect the results presented, see later comments.

Response:

Thank you for this comment. We have edited the Results section of the abstract which now reads as follows and we hope this is more consistent with the presented results: "Results: Participants often reported dramatic impact on both activities and their quality of life. Participants were aware of both antibiotic side effects and resistance, but if they had previously been prescribed antibiotics, many patients believed that they would be necessary for the future treatment of sinusitis. Participants used self-help treatments for short and limited periods of time only, and did feel that regular and consistent use was necessary. In the context of the trial, steam inhalation used for recurrent sinusitis was described as acceptable, but is seen as having limited effectiveness. Nasal irrigation was viewed as acceptable and beneficial by more patients. However, some participants reported that they would not use the treatment again due to the uncomfortable side-effects they experienced, which outweighed any symptom relief which may have resulted had they continued."

Comment:

2. It should be made clear in the discussion which are views of the participants and which are findings from the main trial.

Response:

Thank you for this comment, this section of the abstract has been edited for clarity and now reads as follows: "Participants often reported dramatic impact on both activities and their quality of life. Participants were aware of both antibiotic side effects and resistance, but if they had previously been prescribed antibiotics, many patients believed that they would be necessary for the future treatment of sinusitis. Participants used self-help treatments for short and limited periods of time only, and did feel that regular and consistent use was necessary. In the context of the trial, steam inhalation used for recurrent sinusitis was described as acceptable, but is seen as having limited effectiveness. Nasal irrigation was viewed as acceptable and beneficial by more patients. However, some participants reported that they would not use the treatment again due to the uncomfortable side-effects they experienced, which outweighed any symptom relief which may have resulted had they continued."

Comment:

3. The first strength listed under strengths and limitations is not very clear. I would have thought the novelty of the study was exploring perceptions of particular management strategies? But if it is patients being managed in primary care this could be re-worded.

Response:

Thank you for the comment. It has been edited and now reads as follows: The interviews permitted an exploration of participants' perceptions of particular management strategies for recurrent or chronic sinusitis."

Background

1. It would be helpful to clarify that antibiotics may be beneficial for some patients with (rhino)sinusitis and give examples of "at risk criteria" mentioned in the NICE guidelines.

Response:

Thank you, this has been edited now reads as follows on page 3: "Whilst antibiotics may be beneficial for some patients with sinusitis, the NICE guidelines recommend that antibiotics should not be prescribed for RTIs in most instances, unless a patient meets a specific at-risk criteria, such as being systematically unwell, or at high risk of serious complications due to a pre-existing comorbidity[1]."

Comment:

2. It would be useful to have definitions of recurrent and chronic sinusitis for a non-clinical reader to distinguish whether these are substantially different.

Response:

We agree this should be included. We have added a definition on page 4 as follows: "Participants had had at least two episodes of acute sinusitis or one episode of chronic sinusitis (lasting for 12 weeks or more) in the three years before enrolment[12]."

Comment:

3. It would be useful to specify what participants were required to do as part of the trial. How long were they trying the self-management treatments for? What instructions were they given about using treatments? Did any patients get antibiotics as well? Did all patients only try one of the two strategies? (e.g. no patient had experience of both?)

Response:

We agree that this information should have been included. It has been added to page 4.

Participants were randomly assigned to an advice strategy: to receive advice to use nasal saline irrigation daily, or no such advice; to receive advice to use steam inhalation daily, or no such advice. A combination group included participants randomly assigned to receive advice to perform daily nasal irrigation and daily steam inhalation. This advice is shown in Table 1. All study participants had access to usual care.

Advice strategy Definition

Nasal saline irrigation Participants given verbal instructions and a link to a demonstration video on

YouTube. Participants provided with a neti pot and asked to irrigate their nose daily for six months.
Steam inhalation Participants were asked to inhale steam for five minutes each day.
Usual care Use of medications or referral was at discretion of patient's physician.

Comment:

Methods

1. Recruitment should mention that participants who had withdrawn from the trial were also eligible to take part in interviews.

Response:

Thank you for this comment. It has been edited and now reads as follows: "Participants were recruited from a sample of patients with chronic or recurrent sinusitis who were taking part in the SNIFS trial. Patients had seen the GP for previous episodes of sinusitis, and were also experiencing symptoms currently. Participants in the trial consented to being invited by telephone for an interview. Participants who had withdrawn from the trial were also eligible to take part in the interviews."

Comment:

2. Exact interview lengths should be reported under the results.

Response:

Unfortunately we do not have the exact length of individual interviews. We had ethical approval for interviews to last up to 60 minutes. Interviews were discussed during frequent team meetings and no interview exceeded this limit. We have edited the text on page 5 as follows: "Trained interviewers (LM, SP, BH, CS) conducted telephone interviews (in order to include a wide geographical area), with each lasting approximately half an hour. Ethical approval was in place for interviews to last up to 60 minutes and no interviews exceeded this."

Comment:

3. It would be useful to include a copy of the interview guide to compare this to the themes identified.

Response:

Thank you for this observation, and we acknowledge this should have been included originally. It is now Appendix 1.

Comment:

4. The description of identifying themes and codes in the analysis does not sounds like an inductive process as it sounds like themes were identified straight away?

Response:

We think the reviewer is referring to the opening sentence in the analysis section. We agree this may suggest a more deductive approach. We were open to themes outside of the core aims of the study and it was an inductive process. We have edited this sentence in the Analysis section on page 5 which now reads as follows: "Inductive thematic analysis[14] was conducted on all transcripts to

determine factors that influence patients' decision to consult a GP or use an alternative treatment for sinusitis, as well as being open to themes outside of the core aims of the study."

Comment:

5. Ethical approval is not reported.

Response:

We apologise for this as it should have been reported originally. This has now been included on page 17.

The following three comments have been addressed together:

Results

1. Results do not report the response rate for participants invited to take part in the study.
2. Participant characteristics should include the trial arm to which the participant was randomised and whether they completed trial activities (some withdrew – reasons for this should be noted).
3. Results do not specify whether participants had recurrent or chronic sinusitis (if these are different) and how long they had had the condition. It would also be useful to know how many had treated their symptoms with antibiotics before and how many had used self-management outside of the trial.

Response:

Due to study procedure limitations, the team were unable to collect characteristics of all participants. This has limited our ability to analyse data according to key characteristics, such as trial arm, in comparative ways. However, we do not feel that this detracts much from the value of this research project as a whole, the relevance of its focus, and the need for publication. We have added this as a limitation of the study which reads as follows: "Due to study procedure limitations the team was unable to collect characteristics of all participants and this has reduced our ability to analyse data according to key characteristics, such as trial arm, in comparative ways."

Comment:

4. Did any of these participants have specific "at risk criteria" mentioned in the background which would indicate that antibiotics may be useful for treating their condition – presumably these people were excluded from the trial? Clarify this to establish that these patients can be treated with antibiotics.

Response:

Thank you for this observation, we have edited the Participants and Procedure section on page 4 which now reads as follows: "Participants were recruited from patients with chronic or recurrent sinusitis who were taking part in the SNIFS trial. Patients had seen the GP for previous episodes of sinusitis, and were also experiencing symptoms currently, although they did not have any 'at risk criteria' as this would have excluded them from participating in the trial."

Comment:

5. It is not clear why some words within quotes are presented in bold text.

Response:

We have removed the bold format in all the quotations.

Comment:

6. The subtheme “acceptance of GPs advice” is important and suggests that patients are happy to try self-management strategies if their GP thinks it would help. The acceptance of non-antibiotic strategies following previous experience of antibiotic treatment and beliefs about the effectiveness of antibiotics for sinusitis is unclear and more evidence under this theme would be helpful. At present these results contrast with the subtheme “previous experience of antibiotics”.

Response:

Thank you for this comment, we have edited these sections for clarity and they now read as follows on pages 8 and 9 respectively: “Overwhelmingly, most participants reported that they were happy to accept and follow their GP’s treatment advice. This advice could include taking antibiotics or using various self-help treatments such as nasal irrigation or steam inhalation.”

“Patients who attributed symptom resolution specifically to antibiotics believed they would be effective and might want to take them again for the same symptoms.”

Comment:

7. The subtheme “beneficial but only for severe symptoms” does not seem to be supported by the quotes. The quotes indicate that participants would not bother to try steam inhalation unless their symptoms were “bad” indicating that steam inhalation is seen as effortful and only worth trying if symptoms are disruptive? Neither quote talks about perceived effectiveness.

Response:

This section has been merged with the previous subtheme and on page 11 has been rewritten as follows: “Some participants who had experienced steam inhalation reported that they would only use it if their symptoms were significantly disruptive, positioning steam inhalation as a self-management technique as perhaps better suited for more severe symptoms, with little effect if symptoms were only mild.

“I haven’t used it (steam inhalation) this time, I suppose it’s (sinusitis) not bad enough to use that, but if I couldn’t sleep and that, then I would have to do that (inhale steam)” (Participant L04).

“I only do it (steam inhalation) if I’m feeling really bad, if it’s getting really bad” (Participant L08)”

Comment:

Discussion

1. The results do not support the statement that patients viewed their sinusitis as chronic – need to provide evidence of this or else delete this sentence.

Response:

Thank you for this comment, it has been edited on page 12 and now reads as follows: “We are aware of no prior qualitative studies of patients’ perceptions of steam inhalation and nasal irrigation for recurrent or chronic sinusitis in a primary care setting.”

Comment:

2. Authors report that patients based treatment decisions on past experience – but this counteracts the subtheme about accepting GP advice. Again it is not clear how perceptions of antibiotic effectiveness and previous experience of antibiotics influenced perceptions of using self-management strategies.

Response:

Thank you for this observation and we have edited this section for clarity on page 12: “Patients with recurrent or chronic sinusitis described the often dramatic impact on their activities and quality of life, and viewed their sinusitis as a chronic condition. They based most treatment decisions on past experiences of managing symptoms, although many were willing to accept GP advice.”

Comment:

3. Line 44, reports that patients would be happy not to take antibiotics following GPs advice but evidence for this is not supported in the results.

Response:

We have edited this sentence so it is clearer, as only some, not all, participants, would be happy not to take antibiotics. This sentence is on page 12: “They based most treatment decisions on past experiences of managing symptoms, although many were willing to accept GP advice.”

Comment:

4. The section on comparison with existing literature could be condensed to the main points relevant to the results of this study.

Response:

We have followed your advice and condensed this section over pages 13 and 14: “If participants had been prescribed antibiotics in the past for the treatment of sinusitis, they attributed symptom benefit to antibiotics and accounts indicated beliefs that this treatment would, therefore, be the optimal approach for future management of the condition. In addition, participants also reported a strong acceptance of any GP advice, also suggesting that if GPs prescribe antibiotics for sinusitis, patients’ beliefs that antibiotics may be needed in the future will be strongly reinforced. These findings are supported by previous quantitative research which has documented that prescribing antibiotics for RTIs directly influences patients’ views relating to the need to consult a GP and take antibiotics for future RTIs[15].

One factor maintaining high prescription rates for sinusitis (with more than 90% of individuals receiving antibiotics[2]) is likely to be the significant impact of the condition on the patient’s quality of life – described in this study in such dramatic terms as ‘horrible’ and ‘it literally stops me from having a life’. The significance of impact on quality of life is supported by similar findings from a secondary care sample[11]. A further factor is that sinusitis is second only to chest infections in a long duration of each attack – on average just short of three weeks[16]. Thus, given an unpleasant and long lasting condition, if GPs have nothing else to recommend antibiotics are likely to be used.”

Comment:

5. Implications regarding steam inhalation could be more clearly presented with appropriate weight given to the trial results. Implication for nasal irrigation could be shorter (avoid repeating the results) and it would be helpful to clarify that clinicians are recommended to give greater support to patients in using the technique given the effect shown in the trial

Response:

Thank you for this comment, we have edited this section and hope that it is now more clearly presented on pages 14 and 15: "Patients who had experienced nasal irrigation could be identified in one of two sub-groups: those who described irrigation as an acceptable technique which could relieve their symptoms; and those who described the discomfort experienced during irrigation with limited justification for its use as a treatment. Patients who reportedly found irrigation to be acceptable often reported that they had persisted with it despite initially finding the treatment very uncomfortable, and many reported how they had 'got used to' the discomfort in order to experience the benefits. Therefore, it is possible that the patients who found irrigation unacceptable may have changed their mind if they had persisted with the treatment. In addition, patients who reported irrigation as being unacceptable tended to report side-effects such as water going down their throat. In short, findings suggest that nasal irrigation can be viewed as an acceptable treatment for the symptoms of sinusitis. An important caveat to this, however, is the need for detailed and clear patient information on the correct procedure and the potential benefits of persisting with the technique in terms of finding it easier with increased practice/use of nasal irrigation. Clinicians are recommended to give greater support to patients in using the technique, given the effect shown in the trial (modest benefit at 3 and 6 months from nasal irrigation)."

Comment:

6. In the conclusion, again specify what is a result of the trial and what is participants' views.

Response:

We have edited this section on page 16 so it is now clearer: "The findings from this qualitative study suggest that steam inhalation is viewed as an acceptable treatment which patients are happy to use, although many perceive it as having limited short-term benefit. Nasal irrigation is acceptable to many patients in relieving symptoms, but some find it uncomfortable or mildly unpleasant. However, detailed information on the correct procedure and potential benefits of persisting with the technique may increase the acceptability of nasal irrigation in patients who find it initially uncomfortable, a finding supported by the main trial data."

Comment:

7. Boxes on the COREQ form appear to be blank – these should be reported in the manuscript or else have N/A next to them.

Response:

We have edited the COREQ form and uploaded it as part of this resubmission.

Comment:

8. The four authors listed as undertaking interviews in the methods section do not match the initials reported under contributorship.

Response:

Thank you for this observation. This has been corrected in both the methods section and contributors section to be LM, SP, BH and CW

Reviewer 3

Comment:

Congratulations with this clear and interesting article!

Response:

Thank you very much.

Comment:

- I can't see any information on the ethics approval. It is probably part of the ethics approval for the larger trial, but this is not made explicit.

Response:

Thank you for this observation, we apologise as this should have been included in the original submission. It has now been added.

Comment:

I can't find information on the number/motivations of participants who declined participation in the telephone interviews.

Response:

Due to study procedure limitations, the team were unable to collect characteristics of all participants and we do not have all of this information available. This has limited our ability to analyse data according to key characteristics, such as trial arm, in comparative ways. However, we do not feel that this detracts much from the value of this research project as a whole, the relevance of its focus, and the need for publication. We have added this as a limitation of the study which reads as follows: "Due to study procedure limitations the team was unable to collect characteristics of all participants and this has reduced our ability to analyse data according to key characteristics, such as trial arm, in comparative ways."

Comment:

Document COREQ seems to be incomplete.

Response:

We have edited the COREQ form and uploaded it as part of the resubmission process.

Comment:

In the manuscript, p 6, the term 'feelings' is maybe a bit strange, as it generally refers to psychological feelings or emotions. Could the example given be classified under 'sensations'?

Response:

Thank you for this comment. We have edited this sentence on page 7 and hopefully it is now clearer. It now reads as follows: “A number of factors were used as indicators of severity including a variety of signs and symptoms ranging from pain (head, sinuses, face) to pressure (e.g. nasal passages being blocked), noises (from nose) and sensations (around face and head).”

Comment:

The title refers to 'the decision to consult a GP': I am not sure whether this is really the focus of the results.

Response:

We have edited the title. It is now: “Well, it literally stops me from having a life when it’s really bad’: a nested qualitative interview study of patient views on the use of self-management treatments for the management of recurrent sinusitis (SNIFS trial).”

Comment:

It would be helpful to attach the interview guide as an appendix.

Response:

Thank you for this comment, we apologise as this should have been included in the original submission. It is now Appendix 1.

VERSION 2 – REVIEW

REVIEWER	Sarah Tonkin-Crine University of Oxford, UK
	I have previously published papers with the second and last author.
REVIEW RETURNED	03-Jun-2017

GENERAL COMMENTS	The authors have responded to the majority of reviewer comments which has helped improve the clarity of the manuscript. Background  1. Authors should clarify they are talking about acute, self-limiting RTIs and not all RTIs. Even when self-limiting, symptoms for some RTIs can last 3-4 weeks not 1-2 weeks (acute cough – and also sinusitis mentioned later in the manuscript). 2. The authors state that RTIs account for 60% of general practice antibiotic prescriptions – is this correct? Do the authors mean all RTIs, including pneumonia and other serious conditions? 3. It may be clearer and more concise to explain the SNIFS study design as a 2x2 factorial trial if applicable. Methods  1. Authors should specify how the interview guide was tested, with whom and who agreed the final version. 2. Authors should provide a definition of saturation (or appropriate citation) and clarify at which point saturation was achieved. Were any interviews carried out after saturation was deemed to have occurred?
--

	Results 1. The authors should be able to assess which strategy or strategies each participant had tried by going through the interview transcripts. Even if it was not clear which trial arm each participant had been in (which it should be) the authors could still report the number of participants who reported they had tried either technique before.2. The section “treatment duration short and irregular” could be better supported by an additional or alternative quote.3. I still disagree with authors that under theme 5 either quote is saying anything about the effectiveness of steam inhalation for mild symptoms.4. The authors say they don’t know which participants were in which trial arm but under theme 6 they report that “approximately half of participants who had experience nasal irrigation reported that the technique provided relief”. Discussion 1. The main findings state that “participants were well aware of the limitations of antibiotics”. This appears to imply that participants were aware of the limited benefit antibiotics could provide in treating their sinusitis symptoms. This is not reported in the results. Participants showed some awareness of antibiotic resistance and potential side effects of taking antibiotics, these are both consequences of taking antibiotics rather than limitations. This also links to the third paragraph of comparison with existing literature.2. The main findings state that if patients “had previously been prescribed antibiotics, they often believed they would be necessary for the future treatment of sinusitis”. I am not sure this is supported by the results. Quotes talk about the effectiveness of antibiotics in treating symptoms previously but do not talk about expectations for antibiotics for future symptoms. It is also difficult to understand how many participants had previous experience of taking antibiotics for their condition and how much they were willing to try self-management techniques. This also links to the first paragraph of “comparison with existing literature”. The previous research cited here does not appear to be clearly linked to the present results.3. The abstract states “Participants used self-help treatments for short and limited periods of time only, and did feel that regular and consistent use was necessary”. This disagrees with the main findings and I think a “not” is missing from the sentence?4. As comment above about theme 5 I do not believe the results support the statement “and paradoxically some participants also believed [steam inhalation] was only helpful for the most severe symptoms”.5. The main findings state that some participants who tried nasal irrigation “would not use the treatment again due to the balance of uncomfortable or unpleasant side-effects which outweighed any symptom relief”. It is not clear from the results whether these participants experienced any symptom relief from nasal irrigation or whether they did not try the technique (correctly) for long enough. It may be better to rephrase the sentence as “which outweighed any potential symptom relief”?6. Authors state “patients were aware of the limitations of antibiotics and many would be happy to accept GP advice not to take them, if this was recommended”. This is not supported by the results presented. Previous research cited in this section was not with patients with sinusitis but with LRTI and sore throat, this should be made clear.7. The abstract should be reviewed in light of any further changes made to the discussion, particularly the main findings. Other
--	---

	1. Some boxes on the COREQ form have N/A entered for not applicable which is not correct.
--	---

VERSION 2 – AUTHOR RESPONSE

SNIFS Revision 2 – Response to Reviewers

Comment:

The authors have responded to the majority of reviewer comments which has helped improve the clarity of the manuscript.

Response:

Thank you. We are glad that you feel the clarity of the manuscript has been improved.

Background

Comment:

1. Authors should clarify they are talking about acute, self-limiting RTIs and not all RTIs. Even when self-limiting, symptoms for some RTIs can last 3-4 weeks not 1-2 weeks (acute cough – and also sinusitis mentioned later in the manuscript).

Response:

We have edited the first sentence in the Background section on page 3, which now reads as follows: "Many respiratory tract infections (RTIs) are acute and self-limiting conditions, lasting for around one to two weeks." We hope this shows greater clarity. We are happy to edit further if requested.

Comment:

2. The authors state that RTIs account for 60% of general practice antibiotic prescriptions – is this correct? Do the authors mean all RTIs, including pneumonia and other serious conditions?

Response:

The reference we use for this statement (Tan T, Little P, Stokes T: Antibiotic prescribing for self limiting respiratory tract infections in primary care: summary of NICE guidance. BMJ. 2008, 337:a437) does not specify what this does or does not include, referring to just 'respiratory tract infections'. The reference they cite (Lindbaek M: Prescribing antibiotics to patients with acute cough and otitis media. Br J Gen Pract. 2006,56(524), 164-65) also refers to 'RTIs'. As such we presume this statement to be correct.

Comment:

3. It may be clearer and more concise to explain the SNIFS study design as a 2x2 factorial trial if applicable.

Response:

Thank you for this observation, we have edited this section on pages 3-4. It now reads as follows:

“The study recruited 871 patients from across 72 practices in the South of England. Participants were randomly assigned to an advice strategy in a 2x2 factorial design: to receive advice to use nasal saline irrigation daily, or no such advice; each of these groups was also randomised to receive advice to use steam inhalation daily, or no such advice.”

Methods

Comment:

1. Authors should specify how the interview guide was tested, with whom and who agreed the final version.

Response:

Thank you for this comment, this information has been added to the ‘Interviews’ section on page 5 and now reads as follows: “The guide was developed as part of a student project and tested with a member of staff: this provided a training opportunity for interviewing and feedback on functionality. Previous literature was reviewed and questions thought to be relevant included. It was then refined and agreed by the research team to ensure our aims were met. It included key topic areas while also providing flexibility to explore unanticipated issues. .”

Comment:

2. Authors should provide a definition of saturation (or appropriate citation) and clarify at which point saturation was achieved. Were any interviews carried out after saturation was deemed to have occurred?

Response:

Thank you for this comment. We have edited the end of the ‘Analysis’ section so it now reads as follows: “Data saturation[16] was achieved and recruitment ceased, with no further interviews conducted.”

Results

Comment:

1. The authors should be able to assess which strategy or strategies each participant had tried by going through the interview transcripts. Even if it was not clear which trial arm each participant had been in (which it should be) the authors could still report the number of participants who reported they had tried either technique before.

Response:

Thank you for this comment. This was originally a research project for medical students, who graduated before overall project completion. This, combined with staff turnover, has resulted in the handover of information not being as thorough as we would have ideally liked. Unfortunately, we did not systematically collect the data to answer these questions. Whilst having incomplete information on patient characteristics, such as which strategies each participant had tried, we still believe that this study provides novel and relevant findings. Such findings are relevant to the ongoing development of knowledge surrounding the use of self-management strategies and antibiotics within primary care.

Comment:

2. The section “treatment duration short and irregular” could be better supported by an additional or alternative quote.

Response:

Thank you for this comment. We have added in an additional quote and edited the first sentence for clarity. It now reads as follows:

“Some participants reported that in general they used self-help treatments only for short and limited periods of time. Self-help treatments appeared to be used in an irregular and inconsistent way (e.g. stopping treatment (such as steam inhalation) as soon as relief is first experienced).

Sometimes I think perhaps I’ve – that is excessive [using nasal irrigation twice a day], especially now that I feel everything is clearer, but particularly to start with, that was quite a relief’ (B06). Well I’ve never used it (steam inhalation) more than probably half a dozen times for one session (illness) over a period of days” (Participant L09)”

Comment:

3. I still disagree with authors that under theme 5 either quote is saying anything about the effectiveness of steam inhalation for mild symptoms.

Response:

We have removed the sentence regarding steam inhalation and its effectiveness for mild symptoms. This theme retains its focus on steam inhalation being beneficial, but for short-term relief.

Comment:

4. The authors say they don’t know which participants were in which trial arm but under theme 6 they report that “approximately half of participants who had experience nasal irrigation reported that the technique provided relief”.

Response:

Thank you for this comment, we have edited it as we do not have the exact numbers. We hope this is now more transparent: “Some of the participants who had experienced nasal irrigation reported that the technique provided relief for their symptoms and that although the treatment may be slightly uncomfortable at times, the benefits received outweighed any discomfort suffered.”

Discussion

Comment:

1. The main findings state that “participants were well aware of the limitations of antibiotics”. This appears to imply that participants were aware of the limited benefit antibiotics could provide in treating their sinusitis symptoms. This is not reported in the results. Participants showed some awareness of antibiotic resistance and potential side effects of taking antibiotics, these are both consequences of taking antibiotics rather than limitations. This also links to the third paragraph of comparison with existing literature.

Response:

Thank you for this comment, we agree that this does refer more to consequences of taking antibiotics rather than limitations, and we have made changes as appropriate. The third sentence of the 'Main findings' section on page 12 now reads as follows: "Thus, although many participants were well aware of the potential negative consequences of antibiotics and some did not expect to receive them every time they consulted a GP, if they had previously been prescribed antibiotics, they often believed they would be necessary for the future treatment of sinusitis." On page 13 at the beginning of the third paragraph for 'Comparison with existing literature' 'limitations' has been changed to 'consequences'.

Comment:

2. The main findings state that if patients "had previously been prescribed antibiotics, they often believed they would be necessary for the future treatment of sinusitis". I am not sure this is supported by the results. Quotes talk about the effectiveness of antibiotics in treating symptoms previously but do not talk about expectations for antibiotics for future symptoms.

It is also difficult to understand how many participants had previous experience of taking antibiotics for their condition and how much they were willing to try self-management techniques.

This also links to the first paragraph of "comparison with existing literature". The previous research cited here does not appear to be clearly linked to the present results.

Response:

Thank you for this comment, we have included additional quotations within this section which we hope add clarity as to how participants perceived antibiotics. Past experience is a key driver. The two new quotations (at the end of Theme 3 on page 9) show future intent, and imply a belief in the need for antibiotics if similar symptoms were to arise in the future based on prior experience. "I've got to have antibiotics" (B05) and "Well, I got to the point where I just used to ring them up and say, "I know what I've got – can I have some antibiotics please?" (B04). We hope that these quotes also help with demonstrating the link

Unfortunately, due to handover not being as thorough as we would have liked, we do not have the information available to show precisely how many participants had previously used antibiotics.

Comment:

3. The abstract states "Participants used self-help treatments for short and limited periods of time only, and did feel that regular and consistent use was necessary". This disagrees with the main findings and I think a "not" is missing from the sentence?

Response:

Thank you for this observation. We have edited the abstract (please see Comment 7, below) and this sentence has been removed.

Comment:

4. As comment above about theme 5 I do not believe the results support the statement "and paradoxically some participants also believed [steam inhalation] was only helpful for the most severe symptoms".

Response:

Thank you for this comment. We have edited this section and it now reads as follows which we believe is in keeping with theme 5: "It was commonly perceived as a method which was only beneficial in providing short term relief, and some participants believed it was mostly helpful for severe symptoms."

Comment:

5. The main findings state that some participants who tried nasal irrigation "would not use the treatment again due to the balance of uncomfortable or unpleasant side-effects which outweighed any symptom relief". It is not clear from the results whether these participants experienced any symptom relief from nasal irrigation or whether they did not try the technique (correctly) for long enough. It may be better to rephrase the sentence as "which outweighed any potential symptom relief"?

Response:

Thank you for this comment. We have edited the sentence as suggested.

Comment:

6. Authors state "patients were aware of the limitations of antibiotics and many would be happy to accept GP advice not to take them, if this was recommended". This is not supported by the results presented. Previous research cited in this section was not with patients with sinusitis but with LRTI and sore throat, this should be made clear.

Response:

We say under Theme 2 that most participants reported they would be happy to accept their GP's treatment advice, which could include taking antibiotics or using self-help methods.

We have amended this sentence so it now reads as follows on p13: "This lack of expectation for antibiotics has also been reported in research with patients with LRTI and sore throat, and in particular often contrasts with GP perceptions of high levels of patient pressure to prescribe [e.g. references 15, 18]."

Comment:

7. The abstract should be reviewed in light of any further changes made to the discussion, particularly the main findings.

Response:

Thank you for this observation. We had edited the Results section of the abstract, and it now reads as follows: "Participants often reported dramatic impact on both activities and their quality of life. Participants were aware of both antibiotic side effects and resistance, but if they had previously been prescribed antibiotics, many patients believed that they would be necessary for the future treatment of sinusitis. Participants used self-help treatments for short and limited periods of time only. In the context of the trial, steam inhalation used for recurrent sinusitis was described as acceptable, but is seen as having limited effectiveness. Nasal irrigation was viewed as acceptable and beneficial by more patients. However, some participants reported that they would not use the treatment again due to the uncomfortable side-effects they experienced, which outweighed any symptom relief which may

have resulted had they continued.”

Other

Comment:

1. Some boxes on the COREQ form have N/A entered for not applicable which is not correct.

Response:

We have edited the COREQ form for this submission, and have changed our N/As to ‘no’ if we have not addressed the guide question. However, the COREQ form says to note N/A in the box if not included in the manuscript, hence why we originally completed it as such. We are happy to follow future guidance as to how best to complete this form.

VERSION 3 – REVIEW

REVIEWER	Sarah Tonkin-Crine University of Oxford, UK I have previously published work with G Leydon and P Little
REVIEW RETURNED	17-Jul-2017

GENERAL COMMENTS	The authors have responded to the previous comments and I recommend the manuscript is accepted.
---